# Comparison of Nine Early Warning Scores for Identification of Short-Term Mortality in Acute Neurological Disease in Emergency Department

**DOI:** 10.3390/jpm12040630

**Published:** 2022-04-14

**Authors:** Carlos Durantez-Fernández, Begoña Polonio-López, José L. Martín-Conty, Clara Maestre-Miquel, Antonio Viñuela, Raúl López-Izquierdo, Laura Mordillo-Mateos, Cristina Jorge-Soto, Martín Otero-Agra, Michele Dileone, Joseba Rabanales-Sotos, Francisco Martín-Rodríguez

**Affiliations:** 1Department of Nursing, Physiotherapy and Occupational Therapy, Faculty of Health Sciences, University of Castilla-La Mancha, 45600 Talavera de la Reina, Spain; carlos.durantez@uclm.es (C.D.-F.); begona.polonio@uclm.es (B.P.-L.); clara.maestre@uclm.es (C.M.-M.); antonio.vinuela@uclm.es (A.V.); laura.mordillo@uclm.es (L.M.-M.); 2Technological Innovation Applied to Health Research Group (ITAS), Faculty of Health Sciences, University of Castilla-La Mancha, 45600 Talavera de la Reina, Spain; dileone.michele@uclm.es; 3Department of Emergency, Hospital Universitario Rio Hortega, 47012 Valladolid, Spain; rlopeziz@saludcastillayleon.es; 4Faculty of Medicine, Universidad de Valladolid, 47005 Valladolid, Spain; fmartin@saludcastillayleon.es; 5Prehospital Early Warning Scoring-System Investigation Group, 47005 Valladolid, Spain; 6Faculty of Nursing, University of Santiago de Compostela, 15782 Santiago de Compostela, Spain; cristina.jorge@usc.es; 7SICRUS Research Group, University of Santiago de Compostela, 15782 Santiago de Compostela, Spain; 8CLINURSID Research Group, University of Santiago de Compostela, 15782 Santiago de Compostela, Spain; 9University School of Nursing, University of Vigo, 36001 Pontevedra, Spain; martinoteroagra@gmail.com; 10REMOSS Research Group, Faculty of Education and Sport Sciences, University of Vigo, 36005 Pontevedra, Spain; 11Hospital Virgen del Puerto, Servicio Extremeño de Salud (SES), 10600 Plasencia, Spain; 12Department of Nursing, Physiotherapy and Occupational Therapy, Faculty of Nursing, University of Castilla-La Mancha, 02006 Albacete, Spain; joseba.rabanales@uclm.es; 13Advanced Life Support, Emergency Medical Services (SACYL), 47007 Valladolid, Spain

**Keywords:** critical care, diagnostic accuracy, Early Warning Scores, emergency department, mortality

## Abstract

(1) Background: The aim was screening the performance of nine Early Warning Scores (EWS), to identify patients at high-risk of premature impairment and to detect intensive care unit (ICU) admissions, as well as to track the 2-, 7-, 14-, and 28-day mortality in a cohort of patients diagnosed with an acute neurological condition. (2) Methods: We conducted a prospective, longitudinal, observational study, calculating the EWS [Modified Early Warning Score (MEWS), National Early Warning Score (NEWS), VitalPAC Early Warning Score (ViEWS), Modified Rapid Emergency Medicine Score (MREMS), Early Warning Score (EWS), Hamilton Early Warning Score (HEWS), Standardised Early Warning Score (SEWS), WHO Prognostic Scored System (WPSS), and Rapid Acute Physiology Score (RAPS)] upon the arrival of patients to the emergency department. (3) Results: In all, 1160 patients were included: 808 patients were hospitalized, 199 cases (17%) required ICU care, and 6% of patients died (64 cases) within 2 days, which rose to 16% (183 cases) within 28 days. The highest area under the curve for predicting the need for ICU admissions was obtained by RAPS and MEWS. For predicting mortality, MREMS obtained the best scores for 2- and 28-day mortality. (4) Conclusions: This is the first study to explore whether several EWS accurately identify the risk of ICU admissions and mortality, at different time points, in patients with acute neurological disorders. Every score analyzed obtained good results, but it is suggested that the use of RAPS, MEWS, and MREMS should be preferred in the acute setting, for patients with neurological impairment.

## 1. Introduction

Acute neurological diseases were about 10% to 20% of the medical admissions at Emergency Departments (ED) before the coronavirus disease (COVID-19) [1,2,3], accounting for the third cause of admissions after cardiovascular and respiratory diseases [4]. Common neurological diseases include a variety of major disorders such as stroke, headache, and seizures [2,3,4]; traumatic brain injuries exhibit a significant incidence as well [5].

The reorganization of medical assistance in consequence of the effort focused on the ongoing COVID-19 pandemic, together with the reduction in patients, who are avoiding the hospital, have entailed a decrease in interventions related with neurological problems in many countries [6,7]. These indirect effects of the COVID-19 pandemic have increased the risk of mortality of these neurological diseases, which have previously shown a high mortality risk worldwide (6.55 million deaths by stroke in 2019) [8,9], with a subsequent overload of ED and Intensive Care Units (ICU).

Until now, many studies have focused on possible strategies to decrease the crowding of ED and ICUs. In this context, it was recently demonstrated that the implementation of assistance protocols in ED with the use of Early Warning Scores (EWS) could help to reduce absolute in-hospital mortality rate [10].

When the focus was moved to neurological diseases, particularly acute cerebrovascular diseases, it was demonstrated that a model including age, stroke severity, Early Warning Score, and Performance Status was found to be valid to predict one-year mortality [11]. Furthermore, the onset severity of neurological diseases could be misleading (many clinical patterns are similar and confounding at onset), and a part of those could present an unexpected rapid clinical deterioration. Keeping in mind this aspect, it appears crucial to develop and fix strategies to better predict not only mortality but also the risk of rapid clinical deterioration in neurological diseases.

Protocol assistance for a coordinated response is necessary to assess the severity of the prognostic and the best initial decision [12,13], which certainly involves the use of Early Warning Scores (EWS).

Most of the studies use only one score to evaluate acute neurological disorders (such as stroke [14,15,16,17,18], traumatic brain injury [5,19], acute hypertensive intracerebral haemorrhage [20], or patients with medium and long-term deterioration [20,21]). A wide variety of EWS is now available, and it is crucial to determine which are the best ones for each pathology, from the clinical context and from the phenomenon that is wanted to be predicted [12,13].

To the best of our knowledge, until now no studies performed a direct comparison between the different EWS. The aim of the study was screening the performance of nine Early Warning Scores to identify patients at high-risk of premature impairment and to detect intensive care unit-admissions, as well as to track the 2-, 7-, 14-, and 28-day mortality in a cohort of patients diagnosed with an acute neurological condition.

## 2. Materials and Methods

### 2.1. Study Design and Setting

We conducted a prospective, longitudinal, observational study with adult patients transferred with high priority by ambulance to the EDs of five hospitals of the Health Public System of Castilla y León (Spain) between 1 January 2019 and 30 August 2021. The study conforms to the broader EQUATOR (Improving the Quality and Transparency of Health Research) guidelines [11]

The Research Ethics Committee of all participating centers (Burgos University Hospital, Segovia Hospital Complex, Salamanca University Assistance Complex, Rio Hortega University Hospital and Valladolid University Clinic) have approved this study (Ref. CEIC 2049, MBCA/dgc, PI 18-895, Ref. CEIm PI010-18, PI 2018 10-119).

### 2.2. Population

The sample was selected from all patients over 18 years of age transferred with high priority by ambulance to the EDs of the selected hospitals with a diagnosis of neurological disorders (ischaemic stroke, seizures, haemorrhage, confusion syndrome, degenerative disease, headache, vertigo, tumour, infection, neuromediated syncope, coma). For this study, all patients who were minors or who met any of the following exclusion criteria were excluded: cardiorespiratory arrest, death before or during transfer, pregnancy, discharge in situ, psychiatric pathology, and/or diagnosis of terminal illness.

During the investigation, an attempt was made to minimize the risk of bias by excluding patients without follow-up through the clinical history, from whom the vital signs necessary for the calculation of the scales could not be obtained, or who did not give their informed consent. A family member or legal guardian was contacted to obtain informed consent for patients with an altered clinical condition or level of consciousness.

### 2.3. Outcomes

The principal outcome was the presence of high-risk of premature impairment (intensive care unit-admissions). Secondly, the performance of the scores for 2-, 7-, 14-, and 28-day mortality was evaluated.

### 2.4. Early Warning Scores Selection

For the study, the clinical variables were analyzed on arrival at the patient’s emergency department, which allowed the calculation of the EWS indices, specifically the Modified Early Warning Score (MEWS), National Early Warning Score (NEWS), VitalPAC Early Warning Score (ViEWS), Modified Rapid Emergency Medicine Score (MREMS), Early Warning Score (EWS), Hamilton Early Warning Score (HEWS), Standardised Early Warning Score (SEWS), WHO Prognostic Scored System (WPSS), and Rapid Acute Physiology Score (RAPS).

The MEWS scale was first used as a tool to assess the state of the hospitalized patient; however, its usefulness has spread as a predictor of ICU admissions and predictions of hospital mortality [22,23]. NEWS was categorized by several studies as the most reliable scale to measure a patient’s deterioration at the prehospital level [24,25]. MREMS is a scale derived from REMS (Rapid Emergency Medicine Score), which seeks to predict hospital mortality in real time, adding the variables age and oxygen saturation, of which, for example, RAPS lacks, which comes from the Acute Physiology score and Chronic Health Evaluation II (APACHE-II) and has been shown to be a good rapid prediction tool [26,27]. HEWS [28] was designed to improve the early detection of all hospitalized patients at risk for deterioration, particularly among those with suspected infection. In SEWS, the inclusion of oxygen saturation is shown to have a significant relationship with short- to medium-term mortality [29]. Finally, WPSS can classify patients into five risk groups with different survivals and probabilities evolution [30].

### 2.5. Collection of the Parameters

For these scores, it was necessary to collect information on respiratory rate, oxygen saturation (SpO_2_), need for supplemental oxygen, heart rate, temperature, age, systolic blood pressure (SBP), mean arterial pressure (MAP), and level of consciousness using the Glasgow Coma Scale (GCS) or, alternatively, the AVPU scale (alert, voice, pain, no response).

During triage, the emergency personnel oversaw collecting the clinical and evaluation data to calculate the scores of the different EWS. SpO_2_, SBP, MAP, temperature, and heart rate measurements were obtained using the Connex^®^ Vital Signs Monitor (Welch Allyn, Inc., Skaneateles Falls, NY, USA). Respiratory rate was measured by directly observing the total number of complete respiratory cycles for one minute. The patient’s level of consciousness was assessed using the GCS. All data were included anonymously in an electronic database created for this study.

The rest of the hospital variables (hospitalized patients, ICU admissions, mortality rates, diagnosis of the corresponding group according to the International Classification of Diseases 11th Revision) were collected by an associate researcher from each hospital, by reviewing the electronic medical records 30 days after the index event.

### 2.6. Statistical Analysis

Quantitative variables are presented as a median and interquartile range (IQR), while for categorical variables absolute and relative frequencies were used. Quantitative variables were analyzed using a normality test (Kolmogorov Smirnov) and were compared with a non-parametric Mann–Whitney U test. Effect sizes (ES) were calculated with the Rosenthal r test and classified according to these parameters: Trivial (<0.2); Small (0.2–0.5); Moderate (0.5–0.8); Large (0.8–1.3); Very Large (≥1.3). For categorical variables comparisons, the chi-squared test was used, and ES were calculated with the Cramer V test. For ES classification, these parameters were used: Trivial (<0.1); Small (0.1–0.3); Medium (0.3–0.5); Large (≥0.5).

To compare the predictive capacity of the EWS, the area under the curve (AUC) of the receiver operating characteristic (ROC) was calculated, to predict the need for ICU admission and 2-, 7-, 14-, and 28-day mortality. Using the Youden test, we selected the cut-off point with the best results on each scale for sensitivity, specificity, positive predictive value (PPV), negative predictive value (NPV), positive likelihood ratio, negative likelihood ratio, and diagnostic accuracy (DA). We used the chi-squared test to generate contingency tables that would allow us to establish association relationships through odds ratios. The accuracy of the scores was compared using DeLong’s test.

All analyses were performed with XLSTAT^®^ BioMED software for Microsoft Excel^®^ version 14.4.0 (Microsoft Inc., Redmond, WA, USA) and IBM SPSS Statistics version 20.0 (IBM Corp., Armonk, NY, USA). In all tests, a confidence level of 95% and a *p*-value below 0.05 were considered significant.

## 3. Results

Between 1 January 2019 and 30 August 2021, we recorded a total of 1160 cases of adult patients with neurological disease who were referred to the ED of the five participating hospitals. Figure 1 show the flowchart.

The median age was 71 years (IQR: 53–82 years, range 18–97), and 46% (537 patients) were females. The main reasons for medical check-up were ischaemic stroke (369 cases, 32%) and seizure (282 cases, 24%), and their priority of care according to hospital triage was mainly level II (50%) or level III emergency care (40%). In total, 808 patients were hospitalized, with ICU care required in 199 cases (17%). The mortality of the patient ranged from 6% (64 cases) within 2 days, to 10% (114 cases) within 7 days, to 13% (145 cases) within 14 days, to 15% (173 cases) within 21 days, to 16% (183 cases) within 28 days.

Table 1 and Table 2 compare the clinical variables from initial assessment and the hospital care data of the patients with the variables resulting from this study: the requirement of ICU care and patient mortality at 2-, 7-, 14-, and 28-days.

Considering the clinical deterioration of patients with different types of neurological diseases, we found that the scales with the best capacity to predict the need for ICU care were the RAPS with an AUC of 0.790 (95% CI: 0.751–0.829) and the MEWS with an AUC of 0.789 (95% CI: 0.750–0.828) (Table 3). Regarding the capacity to predict mortality, the MREMS obtained the best score for 2-day mortality with an AUC of 0.929 (95% CI: 0.885–0.973) and for 28-day mortality with an AUC of 0.856 (95% CI: 0.820–0.891) (Table 3).

Table 3 and Appendix A shows the best cut-off values for each EWS, according to sensitivity and specificity (Youden test) for each of the studied events (ICU, 2-, 7-, 14-, 28-day mortality), which allowed to discern their predictive value and diagnostic accuracy. Comparisons of the AUC of the different scoring systems (Appendix A) show that they are significantly different with RAPS and MEWS, having the highest efficacy to predict the ICU admissions and MREMS for 2- and 28-day mortality.

## 4. Discussion

To the best of our knowledge, this is the first study to explore whether several EWS accurately identify the risk of ICU admissions and mortality at different time points in patients with acute neurological disorders. Our findings showed that both RAPS and MEWS performed well. However, regarding mortality prediction, MREMS showed the best capacity to predict it both in the short term and in the long term. Previous studies demonstrated that the use of combined scores, in which EWS was one of the factors included, were able to predict one-year mortality in haemorrhagic and ischaemic stroke patients, with a similar AUC of 85% [11]. Liljehult et al. retrospectively analyzed a cohort of 1113 stroke patients and evaluated the performance of a combined score that included EWS. On the other hand, our total cohort was similar, but we analyzed a not-homogeneous population that included different neurological conditions, so the two studies are not directly comparable. Other studies drew similar conclusions about the usefulness of EWS in mortality and ICU hospitalization prediction in ED, but most are focused on a particular neurological condition as well as traumatic brain injury, brain neoplasm, brain haemorrhage, etc. [19].

Importantly, when applied to neurological disorders, we found a difference in performance of EWS that is mainly based on similar parameters with different relative weight. In this aspect, we must underline that EWS has been developed and used for areas such as ED and ICU, in which the stratification of patients into risk groups helps rapid evaluation, triage, and early management [31,32,33]. Furthermore, EWS can improve the care of critically ill patients, who must be transported from acute settings to hospitals, segregating those with a good prognosis from those with a bad one. Importantly, most of the patients included were diagnosed with ischaemic stroke, epilepsy, and cerebral haemorrhage that included traumatic and not-traumatic etiology, so that our results are mainly affected by these three conditions.

Particularly, in the present cohort, 32% of the patients were diagnosed with stroke, representing the most frequent pathology analyzed. In Spain, stroke care is based on a “Hub and Spoke” system that, together with multimodal imaging, was demonstrated to improve access to revascularization treatment [34]. Revascularization could be achieved mainly by intravenous thrombolysis and/or mechanical thrombectomy: the last one is usually available in Hub Hospitals, but not in Spoke Hospitals, and is indicated for large vessel occlusions [35]. Large vessel occlusion strokes frequently reduce GCS scores, and their general clinical conditions are often worse than those observed in patients with stroke of small vessel occlusions, leading to higher scores in EWS [36]. Keeping in mind this point the use of EWS, particularly of the scales that we demonstrated to have best performances in terms of mortality and ICU risk, could permit to differentiate the transport destiny, bringing “high risk” stroke patients directly to Hub Hospital, reducing treating times and improving the quality of stroke care, permitting a rapid access to advanced imaging and, if necessary, to mechanical thrombectomy. In other words, the use of RAPS, MEWS, and MREMS could help to fasten and ameliorate acute stroke treatment, reducing pre-hospital timings.

It should also be considered that the management of traumatic brain injuries and cerebral haemorrhages depend on the presence of neurosurgery department: interestingly, not all the patients with these conditions are potential candidates for neurosurgery. So, the use of EWS could stratify the patients who are at risk of a rapid clinical deterioration, to anticipate transport to a tertiary hospital (that offers 24 h on-call neurosurgery) or rapid on-site access to neurosurgery, when available.

### Limitations

The major limitations depend on the design and the sample size of the study. Firstly, we conceived and developed an observational study that it is known to increase the risk of a selection bias. However, the fact that several neurological conditions were included could have significantly reduced it. Another important limitation is the sample size: we included more than 1000 patients, but the majority of those were affected by strokes, haemorrhages, and seizures, so that the weight of the final analysis mostly depends on these conditions, hence, de facto reducing our cohort to 50% of the total.

In the sample distribution, according to the need for intensive care of the patients, we found that age groups were significantly different; this situation may be due to confounding bias with older patients, who often have limited ICU care [37,38], despite of the impact of hospitalization in older adults is bigger, especially for polimedication [39]. In total, 90% of the patients admitted to the ICU presented as a base pathology ischaemic stroke, seizures, or haemorrhages. We consider it interesting to carry out future studies to address the predictive capacity of EWS, discriminating between “unplanned ICU admission” and “direct ICU admission from the EDs” in patients with these pathologies. Another limitation consists of the lack of a longer follow-up, which could add interesting information about the presence of residual functional deterioration and long-term mortality. All these limitations did not allow more meaningful conclusions about the role of EWS in stroke patients, and more studies with specific analysis are warranted.

## 5. Conclusions

In conclusion, this study suggests that the use of RAPS, MEWS, and MREMS should be preferred in the acute setting in patients with neurological impairment. High scores indicate worse prognosis and the potential necessity of more intensive and specific treatments.

## Figures and Tables

**Figure 1 jpm-12-00630-f001:**
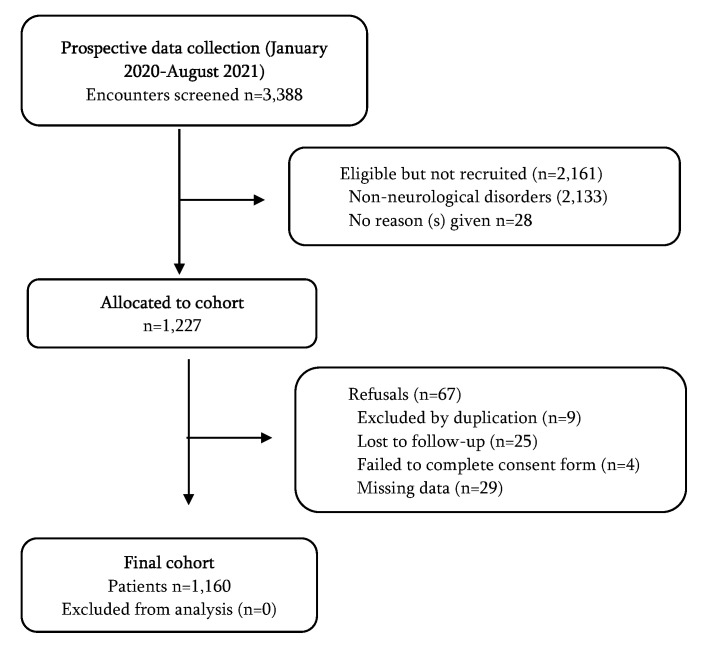
Participant inclusion flow diagram.

**Table 1 jpm-12-00630-t001:** Comparison of patient variables recorded in the emergency department according to the patients’ intensive care.

Variables ^1^	Total	Intensive-Care Unit		*p*-Value and Effect Size ^2^
		Yes	No	
Number	1160 (100%)	199 (17%)	961 (83%)	-
Demographic				
Age (years)	71 (53–82)	64 (53–77)	72 (54–83)	*p* = 0.001 * (0.10) ^T^
Sex				
Male	623 (54%)	119 (60%)	504 (52%)	*p* = 0.06
Female	537 (46%)	80 (40%)	457 (48%)	
Initial evaluation				
Pulse (bpm)	81 (69–93)	83 (70–96)	80 (68–93)	*p* = 0.17
Respiratory rate (bpm)	15 (13–17)	15 (14–16)	14 (13–17)	*p* = 0.025 * (0.07) ^T^
Temperature (°C)	36.0 (35.8–36.5)	36.0 (35.7–36.5)	36.0 (35.8–36.5)	*p* = 0.25
Systolic Blood Pressure (mmHg)	138 (120–158)	138 (117–167)	138 (120–157)	*p* = 0.72
Diastolic Blood Pressure (mmHg)	77 (67–87)	78 (65–93)	77 (67–87)	*p* = 0.39
Mean Blood Pressure (mmHg)	98 (87–109)	99 (85–119)	98 (87–109)	*p* = 0.52
SpO_2_ (%)	97 (95–98)	98 (95–100)	96 (94–98)	*p* < 0.001 * (0.12) ^T^
Air oxygen	303 (26%)	135 (68%)	168 (18%)	*p* < 0.001 * (0.43) ^M^
FiO_2_ (%)	0.21 (0.21–0.24)	0.50 (0.21–0.99)	0.21 (0.21–0.21)	*p* < 0.001 * (0.48) ^S^
Glasgow Coma Scale (total)	15 (12–15)	4 (3–14)	15 (14–15)	*p* < 0.001 * (0.48) ^S^
Eye Opening Response	4 (3–4)	1 (1–3)	4 (4–4)	*p* < 0.001 * (0.48) ^S^
Verbal Response	5 (4–5)	1 (1–5)	5 (5–5)	*p* < 0.001 * (0.49) ^S^
Motor Response	6 (6–6)	2 (1–6)	6 (6–6)	*p* < 0.001 * (0.54) ^M^
Hospital Triage				
Level I: Resuscitation	121 (10%)	93 (47%)	28 (3%)	*p* < 0.001 * (0.54) ^L^
Level II: Emergency	577 (50%)	91 (46%)	486 (51%)	*p* = 0.21
Level III: Urgency	462 (40%)	15 (7%)	447 (46%)	*p* < 0.001 * (0.30) ^M^
Pathology				
Ischaemic stroke	369 (32%)	37 (19%)	332 (35%)	*p* < 0.001 * (0.13) ^S^
Seizures	282 (24%)	29 (14%)	253 (26%)	*p* < 0.001 * (0.10) ^S^
Haemorrhage	204 (18%)	99 (50%)	105 (11%)	*p* < 0.001 * (0.38) ^M^
Confusion syndrome	69 (6%)	4 (2%)	65 (7%)	*p* = 0.010 * (0.08) ^T^
Degenerative disease	66 (6%)	2 (1%)	64 (7%)	*p* = 0.002 * (0.09) ^T^
Headache	42 (3%)	0 (0%)	42 (4%)	*p* = 0.003 * (0.09) ^T^
Vertigo	31 (3%)	0 (0%)	31 (3%)	*p* = 0.010 * (0.08) ^T^
Tumour	30 (3%)	2 (1%)	28 (3%)	*p* = 0.12
Infection	24 (2%)	14 (7%)	10 (1%)	*p* < 0.001 * (0.16) ^S^
Neuromediated syncope	24 (2%)	0 (0%)	24 (2%)	*p* = 0.024 * (0.07) ^T^
Coma	19 (1%)	12 (6%)	7 (1%)	*p* < 0.001 * (0.16) ^S^
Hospital outcomes				
Inpatients	808 (70%)	198 (99%)	610 (64%)	*p* < 0.001 * (0.30) ^M^
Hospitalization days (inpatients)	7 (4–13)	10 (4–20)	7 (4–11)	*p* < 0.001 * (0.14) ^T^
Mortality				
2-day	64 (6%)	38 (19%)	26 (3%)	*p* < 0.001 * (0.27) ^S^
7-day	114 (10%)	54 (27%)	60 (6%)	*p* < 0.001 * (0.27) ^S^
14-day	145 (13%)	65 (33%)	80 (8%)	*p* < 0.001 * (0.28) ^S^
21-day	173 (15%)	79 (40%)	94 (10%)	*p* < 0.001 * (0.32) ^M^
28-day	183 (16%)	84 (42%)	99 (10%)	*p* < 0.001 * (0.33) ^M^
EWS analyzed				
NEWS	4 (2–6)	6 (5–8)	3 (1–5)	*p* < 0.001 * (0.35) ^S^
ViEWS	3 (1–6)	6 (5–8)	3 (1–5)	*p* < 0.001 * (0.35) ^S^
MEWS	2 (1–3)	4 (2–6)	2 (1–3)	*p* < 0.001 * (0.38) ^S^
MREMS	5 (3–7)	8 (5–10)	4 (2–6)	*p* < 0.001 * (0.33) ^S^
EWS	1 (0–3)	3 (2–5)	1 (0–2)	*p* < 0.001 * (0.37) ^S^
HEWS	3 (2–4)	5 (3–6)	3 (1–4)	*p* < 0.001 * (0.30) ^S^
SEWS	1 (0–3)	3 (2–5)	1 (0–2)	*p* < 0.001 * (0.37) ^S^
RAPS	2 (0–4)	4 (2–6)	2 (0–3)	*p* < 0.001 * (0.39) ^S^
WPSS	2 (0–4)	3 (3–6)	2 (0–3)	*p* < 0.001 * (0.29) ^S^

^1^ Values expressed as a total number (fraction) and medians (1st quartile−3rd quartile) as appropriate. Bracketed numbers indicate 95% confidence interval. ^2^ The *p*-values were calculated with the Mann–Whitney U test and chi-squared test. Effect Size was calculated with the Rosenthal r test [Trivial ^(T)^ (<0.2); Small ^(S)^ (0.2–0.5); Moderate ^(M)^ (0.5–0.8); Large ^(L)^ (0.8–1.3) and Cramer V test [Trivial ^(T)^ (<0.1); Small ^(S)^ (0.1–0.3); Medium ^(M)^ (0.3–0.5); Large ^(L)^ ≥0.5]. SpO_2_: Oxygen saturation; NEWS: National Early Warning Score; ViEWS: Vital PAC Early Warning Score; MEWS: Modified Early Warning Score; MREMS: Modified Rapid Emergency Medicine Score; EWS: Early Warning Score; HEWS: Hamilton Early Warning Score; SEWS: Standardised Early Warning Score; RAPS: Rapid Acute Physiology Score; WPSS: WHO Prognostic Scored System. * *p* < 0.05.

**Table 2 jpm-12-00630-t002:** Comparison of patient variables recorded in the emergency department according to 2-, 7-, 14-, and 28-day mortality.

Variables	Survivors	Non-Survivors					*p*-Value
		2-Day	7-Day	14-Day	21-Day	28-Day	
Number	977 (84%)	64 (6%)	114 (10%)	145 (13%)	173 (15%)	183 (16%)	
Demographic							
Age (years)	67 (52–80)	79 (66–84)	80 (69–86)	80 (70–87)	80 (67–86)	79 (67–86)	*p* < 0.001 * (0.23) ^S^
Sex							
Male	528 (54%)	36 (56%)	60 (53%)	70 (48%)	90 (52%)	95 (52%)	*p* = 0.60
Female	449 (46%)	28 (44%)	54 (47%)	75 (52%)	83 (48%)	88 (48%)	
Initial evaluation							
Pulse (bpm)	81 (69–93)	85 (68–100)	81 (68–96)	79 (68–93)	80 (69–94)	81 (69–95)	*p* = 0.70
Respiratory rate (bpm)	14 (13–17)	15 (15–21)	15 (15–19)	15 (15–19)	15 (15–18)	15 (15–18)	*p* < 0.001 * (0.15) ^T^
Temperature (°C)	36.1 (35.8–36.5)	36.0 (35.0–36.7)	36.0 (35.3–36.5)	36.0 (35.4–36.6)	36.0 (35.5–36.6)	36.0 (35.5–36.6)	*p* = 0.017 * (0.07) ^T^
Systolic Blood Pressure (mmHg)	137 (120–156)	147 (107–174)	145 (120–168)	145 (120–170)	144 (123–170)	144 (123–170)	*p* = 0.006 * (0.08) ^T^
Diastolic Blood Pressure (mmHg)	77 (67–87)	76 (60–96)	75 (60–93)	79 (61–93)	80 (64–92)	80 (65–93)	*p* = 0.16
Mean Blood Pressure (mmHg)	97 (87–108)	101 (77–122)	100 (82–121)	100 (83–119)	101 (86–119)	101 (87–119)	*p* = 0.022 * (0.07) ^T^
SpO_2_ (%)	97 (95–98)	96 (91–100)	95 (92–99)	95 (92–99)	96 (93–99)	96 (93–99)	*p* = 0.013 * (0.07) ^T^
Air oxygen	190 (19%)	58 (91%)	85 (75%)	101 (70%)	112 (65%)	113 (62%)	*p* < 0.001 * (0.35) ^M^
FiO_2_ (%)	0.21 (0.21–0.21)	0.50 (0.40–0.99)	0.50 (0.21–0.99)	0.40 (0.21–0.99)	0.40 (0.21–0.99)	0.31 (0.21–0.99)	*p* < 0.001 * (0.38) ^S^
Glasgow Coma Scale	15 (14–15)	3 (3–7)	5 (3–11)	6 (3–12)	7 (3–13)	8 (3–13)	*p* < 0.001 * (0.48) ^S^
Eye Opening Response	4 (3–4)	1 (1–1)	1 (1–3)	1 (1–3)	2 (1–3)	2 (1–3)	*p* < 0.001 * (0.46) ^S^
Verbal Response	5 (5–5)	1 (1–1)	1 (1–3)	1 (1–4)	1 (1–4)	2 (1–4)	*p* < 0.001 * (0.50) ^M^
Motor Response	6 (6–6)	1 (1–3)	3 (1–5)	3 (1–5)	3 (1–6)	4 (1–6)	*p* < 0.001 * (0.50) ^M^
Hospital Triage							
Level I: Resuscitation	63 (6%)	33 (52%)	42 (37%)	48 (33%)	58 (34%)	58 (32%)	*p* < 0.001 * (0.30) ^M^
Level II: Emergency	477 (49%)	26 (40%)	61 (53%)	80 (55%)	92 (53%)	100 (54%)	*p* = 0.15
Level III: Urgency	437 (45%)	5 (8%)	11 (10%)	17 (12%)	23 (13%)	25 (14%)	*p* < 0.001 * (0.23) ^S^
Pathology							
Ischaemic stroke	318 (33%)	7 (11%)	26 (23%)	36 (25%)	46 (26%)	51 (28%)	*p* = 0.21
Seizures	276 (28%)	1 (2%)	2 (2%)	4 (3%)	4 (2%)	6 (3%)	*p* < 0.001 * (0.21) ^S^
Haemorrhage	112 (12%)	42 (65%)	67 (59%)	79 (55%)	90 (52%)	92 (51%)	*p* < 0.001 * (0.37) ^M^
Confusion syndrome	65 (7%)	1 (2%)	1 (1%)	2 (1%)	3 (2%)	4 (2%)	*p* = 0.019 * (0.07) ^T^
Degenerative disease	60 (6%)	1 (2%)	2 (2%)	5 (3%)	6 (4%)	6 (3%)	*p* = 0.13
Headache	42 (4%)	0 (0%)	0 (0%)	0 (0%)	0 (0%)	0 (0%)	*p* = 0.004 * (0.08) ^T^
Vertigo	31 (3%)	0 (0%)	0 (0%)	0 (0%)	0 (0%)	0 (0%)	*p* = 0.015 * (0.07) ^T^
Tumours	24 (2%)	0 (0%)	1 (1%)	3 (2%)	6 (4%)	6 (3%)	*p* = 0.52
Neuromediated syncope	24 (2%)	0 (0%)	0 (0%)	0 (0%)	0 (0%)	0 (0%)	*p* = 0.032 * (0.06) ^T^
Infections	17 (2%)	4 (6%)	5 (4%)	6 (4%)	7 (4%)	7 (4%)	*p* = 0.07
Coma	8 (1%)	8 (12%)	10 (8%)	10 (7%)	11 (6%)	11 (6%)	*p* < 0.001 * (0.15) ^S^
Hospital outcomes							
Inpatients	627 (64%)	63 (98%)	113 (99%)	143 (99%)	171 (99%)	181 (99%)	*p* < 0.001 * (0.28) ^S^
Hospitalization days (inpatients)	8 (5–13)	1 (1–2)	2 (1–4)	3 (1–7)	4 (2–10)	5 (2–11)	*p* < 0.001 * (0.18) ^T^
Intensive care unit	115 (12%)	38 (59%)	54 (47%)	65 (45%)	79 (46%)	84 (46%)	*p* < 0.001 * (0.33) ^M^
EWS analyzed							
NEWS	3 (1–5)	9 (7–11)	7 (6–10)	7 (6–9)	7 (5–9)	7 (5–9)	*p* < 0.001 * (0.40) ^S^
ViEWS	3 (1–5)	8 (6–11)	7 (6–10)	7 (5–9)	7 (5–9)	7 (5–9)	*p* < 0.001 * (0.40) ^S^
MEWS	2 (1–3)	6 (4–7)	5 (4–6)	5 (3–6)	5 (3–6)	4 (3–6)	*p* < 0.001 * (0.41) ^S^
MREMS	4 (2–6)	11 (9–13)	10 (7–11)	9 (7–11)	9 (6–11)	9 (6–11)	*p* < 0.001 * (0.45) ^S^
EWS	1 (0–2)	4 (3–6)	4 (3–5)	4 (3–5)	4 (2–5)	4 (2–5)	*p* < 0.001 * (0.40) ^S^
HEWS	3 (1–4)	6 (5–9)	5 (4–7)	5 (4–6)	5 (3–6)	5 (3–6)	*p* < 0.001 * (0.34) ^S^
SEWS	1 (0–2)	4 (3–6)	4 (3–5)	4 (3–5)	4 (2–5)	4 (2–5)	*p* < 0.001 * (0.40) ^S^
RAPS	2 (0–3)	6 (4–8)	5 (3–7)	5 (3–7)	4 (3–6)	4 (3–6)	*p* < 0.001 * (0.40) ^S^
WPSS	2 (0–3)	5 (3–8)	1 (1–2)	5 (3–6)	4 (3–6)	4 (3–6)	*p* < 0.001 * (0.37) ^S^

^1^ Values expressed as a total number (fraction) and medians (1st quartile−3rd quartile) as appropriate. Bracketed numbers indicate 95% confidence interval. ^2^ The *p*-values were calculated with the Mann–Whitney U test and chi-squared test. Effect Size was calculated with the Rosenthal r test [Trivial ^(T)^ (<0.2); Small ^(S)^ (0.2–0.5); Moderate ^(M)^ (0.5–0.8); Large ^(L)^ (0.8–1.3) and Cramer V test [Trivial ^(T)^ (<0.1); Small ^(S)^ (0.1–0.3); Medium ^(M)^ (0.3–0.5); Large ^(L)^ ≥0.5]. SpO_2_: Oxygen saturation; NEWS: National Early Warning Score; ViEWS: Vital PAC Early Warning Score; MEWS: Modified Early Warning Score; MREMS: Modified Rapid Emergency Medicine Score; EWS: Early Warning Score; HEWS: Hamilton Early Warning Score; SEWS: Standardised Early Warning Score; RAPS: Rapid Acute Physiology Score; WPSS: WHO Prognostic Scored System. * *p* < 0.05.

**Table 3 jpm-12-00630-t003:** AUROC, cut-off points for combined sensitivity and specificity with the best score (Youden’s test) for the different scales analyzed (for intensive care unit, 2-, and 28-day mortality).

Scores		Intensive Care Unit	Non-Survivors 2-Day	Non-Survivors 28-Day
NEWS				
	Cut-off	5	6	5
	AUROC	0.769 (0.728–0.809)	0.908 (0.859–0.957)	0.815 (0.776–0.854)
	Sensitivity	77.4 (71.1–82.6)	93.8 (85.0–97.5)	80.3 (74.6–86.1)
	Specificity	70.1 (67.2–72.9)	75.5 (72.9–78.0)	69.9 (67.0–72.8)
	PPV	34.9 (30.6–39.5)	18.3 (14.5–22.8)	30.1 (27.3–33.0)
	NPV	93.7 (91.7–95.3)	99.5 (98.8–99.8)	19.7 (14.6–26.0)
	Likelihood ratio +	2.59 (2.29–2.93)	3.83 (3.39–4.33)	2.67 (2.37–3.01)
	Likelihood ratio −	0.32 (0.25–0.42)	0.08 (0.03–0.21)	0.28 (0.21–0.38)
	Odds ratio	8.04 (5.61–11.52)	46.34 (16.69–128.71)	9.49 (6.43–14.00)
	Diagnostic accuracy	71.4 (68.7–73.9)	76.6 (74.0–78.9)	71.6 (68.9–74.1)
ViEWS				
	Cut-off	5	5	5
	AUROC	0.768 (0.727–0.808)	0.907 (0.857–0.956)	0.813 (0.774–0.852)
	Sensitivity	75.9 (69.5–81.3)	100.0 (94.3–100.0)	79.2 (72.8–84.5)
	Specificity	72.3 (69.4–75.1)	67.8 (65.0–70.5)	72.2 (69.3–74.9)
	PPV	36.2 (31.7–40.9)	15.3 (12.2–19.1)	34.8 (30.4–39.5)
	NPV	93.5 (91.5–95.1)	100.0 (99.5–100.0)	94.9 (93.1–96.3)
	Likelihood ratio +	2.74 (2.41–3.12)	3.10 (2.85–3.38)	2.85 (2.51–3.23)
	Likelihood ratio −	0.33 (0.26–0.43)	0.00 (0.00–0.00)	0.29 (0.22–0.38)
	Odds ratio	8.22 (5.77–11.71)	-	9.89 (6.74–14.51)
	Diagnostic accuracy	72.9 (70.3–75.4)	69.6 (66.9–72.1)	73.3 (70.7–75.7)
MEWS				
	Cut-off	4	4	3
	AUROC	0.789 (0.750–0.828)	0.914 (0.866–0.961)	0.818 (0.780–0.857)
	Sensitivity	64.8 (58.0–71.1)	92.2 (83.0–96.6)	81.4 (75.2–86.4)
	Specificity	84.3 (81.9–86.5)	79.8 (77.4–82.1)	70.8 (67.9–73.6)
	PPV	46.1 (40.3–51.9)	21.1 (16.7–26.2)	34.3 (30.0–38.9)
	NPV	92.0 (90.1–93.7)	99.4 (98.7–99.8)	95.3 (93.5–96.6)
	Likelihood ratio +	4.13 (3.45–4.93)	4.57 (3.98–5.25)	2.79 (2.48–3.15)
	Likelihood ratio −	0.42 (0.34–0.51)	0.10 (0.04–0.23)	0.26 (0.19–0.36)
	Odds ratio	9.89 (7.04–13.87)	46.72 (18.53–117.79)	10.64 (7.15–15.83)
	Diagnostic accuracy	80.9 (78.6–83.1)	80.5 (78.1–82.7)	72.5 (69.9–75.0)
MREMS				
	Cut-off	8	8	6
	AUROC	0.755 (0.714–0.796)	0.929 (0.885–0.973)	0.856 (0.820–0.891)
	Sensitivity	50.8 (43.9–57.6)	87.5 (77.2–93.5)	83.6 (77.6–88.3)
	Specificity	90.4 (88.4–92.1)	87.5 (85.4–89.3)	70.1 (67.2–72.9)
	PPV	52.3 (45.3–59.3)	29.0 (23.1–35.8)	34.4 (30.1–38.9)
	NPV	89.9 (87.8–91.6)	99.2 (98.4–99.6)	95.8 (94.1–97.0)
	Likelihood ratio +	5.30 (4.18–6.72)	7.00 (5.84–8.40)	2.80 (2.49–3.14)
	Likelihood ratio −	0.54 (0.47–0.63)	0.14 (0.07–0.27)	0.23 (0.17–0.33)
	Odds ratio	9.73 (6.85–13.83)	49.00 (22.87–105.00)	11.96 (7.90–18.11)
	Diagnostic accuracy	83.6 (81.4–85.6)	87.5 (85.5–89.3)	72.2 (69.6–74.7)
EWS				
	Cut-off	3	3	3
	AUROC	0.774 (0.733–0.814)	0.895 (0.843–0.947)	0.810 (0.771–0.850)
	Sensitivity	69.8 (63.1–75.8)	92.2 (83.0–96.6)	72.1 (65.2–78.1)
	Specificity	81.2 (78.6–83.5)	76.2 (73.6–78.6)	80.8 (78.2–83.1)
	PPV	43.4 (38.1–48.9)	18.4 (14.6–23.1)	41.3 (36.0–46.7)
	NPV	92.9 (90.9–94.4)	99.4 (98.6–99.7)	93.9 (92.1–95.4)
	Likelihood ratio +	3.71 (3.16–4.35)	3.87 (3.41–4.40)	3.75 (3.20–4.39)
	Likelihood ratio −	0.37 (0.30–0.46)	0.10 (0.04–0.24)	0.35 (0.27–0.44)
	Odds ratio	9.98 (7.08–14.07)	37.75 (14.99–95.06)	10.86 (7.58–15.57)
	Diagnostic accuracy	79.2 (76.8–81.5)	77.1 (74.6–79.4)	79.4 (77.0–81.6)
HEWS				
	Cut-off	4	5	4
	AUROC	0.728 (0.686–0.771)	0.865 (0.807–0.922)	0.769 (0.727–0.811)
	Sensitivity	68.3 (61.6–74.4)	79.7 (68.3–87.7)	72.1 (65.2–78.1)
	Specificity	70.9 (67.9–73.6)	79.4 (76.9–81.7)	70.9 (68.0–73.7)
	PPV	32.7 (28.4–37.3)	18.4 (14.3–23.4)	31.7 (27.4–36.4)
	NPV	91.5 (89.3–93.3)	98.5 (97.5–99.1)	93.1 (91.1–94.7)
	Likelihood ratio +	2.35 (2.05–2.69)	3.86 (3.26–4.58)	2.48 (2.17–2.83)
	Likelihood ratio −	0.45 (0.36–0.55)	0.26 (0.16–0.42)	0.39 (0.31–0.50)
	Odds ratio	5.25 (3.78–7.30)	15.10 (8.07–28.25)	6.32 (4.44–8.98)
	Diagnostic accuracy	70.4 (67.7–73.0)	79.4 (77.9–81.6)	71.1 (68.4–73.7)
SEWS				
	Cut-off	3	3	3
	AUROC	0.773 (0.733–0.814)	0.895 (0.843–0.947)	0.810 (0.771–0.850)
	Sensitivity	69.8 (63.1–75.8)	92.2 (83.0–96.6)	72.1 (65.2–78.1)
	Specificity	81.2 (78.6–83.5)	76.2 (73.6–78.6)	80.8 (78.2–83.1)
	PPV	43.4 (38.1–48.9)	18.4 (14.6–23.1)	41.3 (36.0–46.7)
	NPV	92.9 (90.9–94.4)	99.4 (98.6–99.7)	93.9 (92.1–95.4)
	Likelihood ratio +	3.71 (3.16–4.35)	3.87 (3.41–4.40)	3.75 (3.20–4.39)
	Likelihood ratio −	0.37 (0.30–0.46)	0.10 (0.04–0.24)	0.35 (0.27–0.44)
	Odds ratio	9.98 (7.08–14.07)	37.75 (14.99–95.06)	10.86 (7.58–15.57)
	Diagnostic accuracy	79.2 (76.8–81.5)	77.1 (74.6–79.4)	79.4 (77.0–81.6)
RAPS				
	Cut-off	4	4	3
	AUROC	0.790 (0.751–0.829)	0.902 (0.852–0.953)	0.806 (0.767–0.846)
	Sensitivity	67.3 (60.5–73.5)	87.5 (77.2–93.5)	77.6 (71.0–83.0)
	Specificity	82.5 (80.0–84.8)	77.6 (75.0–79.9)	72.2 (69.3–74.9)
	PPV	44.4 (38.9–50.0)	18.5 (14.6–23.3)	34.3 (29.9–39.0)
	NPV	92.4 (90.5–94.0)	99.1 (98.2–99.5)	94.5 (92.6–95.9)
	Likelihood ratio +	3.85 (3.26–4.56)	3.90 (3.38–4.50)	2.79 (2.45–3.17)
	Likelihood ratio −	0.40 (0.32–0.49)	0.16 (0.08–0.31)	0.31 (0.24–0.41)
	Odds ratio	9.73 (6.93–13.67)	24.19 (11.38–51.42)	8.98 (6.17–13.06)
	Diagnostic accuracy	79.9 (77.5–82.1)	78.1 (75.6–80.4)	73.0 (70.4–75.5)
WPSS				
	Cut-off	3	3	3
	AUROC	0.716 (0.673–0.759)	0.846 (0.785–0.906)	0.790 (0.749–0.830)
	Sensitivity	83.9 (78.2–88.4)	100.0 (94.3–100.0)	89.6 (84.4–93.3)
	Specificity	58.0 (54.8–61.0)	53.7 (50.8–56.7)	58.3 (55.2–61.4)
	PPV	29.2 (25.7–33.1)	11.2 (8.9–14.1)	28.7 (25.2–32.6)
	NPV	94.6 (92.4–96.1)	100.0 (99.4–100.0)	96.8 (95.0–97.9)
	Likelihood ratio +	2.00 (1.81–2.20)	2.16 (2.03–2.30)	2.15 (1.97–2.35)
	Likelihood ratio −	0.28 (0.20–0.38)	0.00 (0.00–0.00)	0.18 (0.12–0.27)
	Odds ratio	7.20 (4.83–10.73)	-	12.09 (7.39–19.77)
	Diagnostic accuracy	62.4 (59.6–65.2)	56.3 (53.4–59.1)	63.3 (60.5–66.0)

Bracketed numbers indicate 95% confidence interval. NEWS: National Early Warning Score; ViEWS: Vital PAC Early Warning Score; MEWS: Modified Early Warning Score; MREMS: Modified Rapid Emergency Medicine Score; EWS: Early Warning Score; HEWS: Hamilton Early Warning Score; SEWS: Standardised Early Warning Score; RAPS: Rapid Acute Physiology Score; WPSS: WHO Prognostic Scored System; AUROC: area under the receiver operating characteristics; PPV: positive predictive value; NPV: negative predictive value; Likelihood ratio +: positive likelihood ratio; Likelihood ratio −: negative likelihood ratio.

## Data Availability

Not applicable.

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
