# Peer review of "Comparison of Nine Early Warning Scores for Identification of Short-Term Mortality in Acute Neurological Disease in Emergency Department"

_jpm, 2022, doi:10.3390/jpm12040630_

Round 1

Reviewer 1 Report

This manuscript entitled “Comparison of nine Early Warning Scores for identification of short-term mortality in acute neurological disease in Emergency Department” was designed to screen the performance of nine Early Warning Scores (EWS), to identify patients at high-risk of premature impairment, to detect intensive care unit (ICU) admissions, as well as 2-, 7-, 14- and 28-day mortality in a cohort of patients diagnosed with an acute neurological condition. The value to perform this research was well described in Background. The study design was solid, and statistical methods were used where appropriately. However, there are still some flaws or confused statements in this manuscript. 

In conclusion of Abstract and in Discussion, the authors stated that they explored whether several EWS accurately identifies the risk of unplanned-ICU admissions as well 42 as mortality, at different time points, in patients with acute neurological disorders. But in Table 1, the authors showed comparison of patient variables recorded in the emergency department according to the patients’ intensive care, which did not mention “unplanned ICU admission.” In Results, the authors stated that “Considering the clinical deterioration of patients with different types of neurological 204 diseases, we found that the scales with the best capacity to predict the need for ICU care 205 were…” (line 204-205). The various early warning scores were developed primarily to detect the clinical deterioration of patients admitted to general ward of hospital, which may lead to unplanned ICU admission from the ward. The term “unplanned ICU-admission”, or also called “indirect ICU admission”, usually refers to patients who were initially admitted to general wards, and then transferred to ICU after clinical deteriorations. In the study design of this manuscript, the authors aim on the capacity of various early warning scores to predict ICU admission, but it is no clear whether the authors focused on direct ICU admission from the EDs, or indirect ICU admission, or both. For direct ICU-admission from the ED, the calculated early warning scores at triage may not be assumed to have “prediction” capacity, because the decision of ICU admission (usually based on the patients’ clinical conditions and vital signs at triage) are concomitant, rather than a “future” event. However, the application of early warning scores in prehospital setting, as stated by the authors in Discussion, could be still reasonable. 

In Table 4, the authors showed the performances of different scales to predict 7-, 14- and 21-day mortality. The results were similar to those on 2- and 28-day mortality, as shown in Table 3. The results of 2- and 28-day mortality were enough to represent short term and long terms mortality, respectively. Presentation of 7-, 14- and 21-day mortality might be redundant accordingly. The authors may need to describe the role and the value of these results.

Author Response

We wish to thank you all for your constructive comments in this round of review. Your comments provided valuable insights to refine its contents and analysis. In this document, we try to address the issues raised as best as possible.

We appreciate the reviewer's clarification on the differences between “direct ICU admission from EDs” and “indirect ICU admission”. We agree on the needy to clarify the terminology on the ICU admission and also to change the paragraphs indicated by the reviewer along the text. In our study, an associate researcher from each hospital reviewed the electronic medical records 30 days after the index event to collect the remaining hospital variables (inpatients, intensive care unit [ICU] admissions, mortality rates, and diagnosis). So that, we use data on “direct admission to the ICU from the EDs” and “indirect admission to the ICU”.

The changes made to the text are:

Line 42: we replace “the risk of unplanned ICU admissions” with “the risk of ICU admissions”.

Line 113-114: We replaced “unplanned intensive care unit admissions” with 2intensive care unit admissions”.

Line 150-153: The rest of the hospital variables (hospitalized patients, ICU admissions, mortality rates and diagnosis of the corresponding group according to the International Classification of Diseases 11th Revision) were collected by an associate researcher from each hospital by reviewing the electronic medical records 30 days after the index event.

Line 237: we replace “unplanned ICU admissions” with “ICU admissions”

We agree on the reviewer´s comment and also consider that the information included in tables 3 and 4 is especially relevant for understanding the article. We consider it could be appropriate to transfer table 4 to supplementary material.

Reviewer 2 Report

The authors present work comparing prioritization scores for patients going into ICU for neurological disease. 

Here are the issues that can be improved :

  • First of all, there is a difference regarding age between patients in ICU and the others. It should be discussed or even incorporated into the statistical analysis if age is not to be seen as a confounding bias with older patients more often limited in care.
  • As 90% of the patients are represented by ischaemic stroke, seizures and haemorrhage in ICU, it should be interresting to do a sub-group analysis only for these patients.
  • Tables 3 and 4 are too long and difficult to read, although they are important data and should be placed in the supplementary data. However, the AUC, PPV, and NPV of the ROC curve could be noted as a figure in the manuscript with the ROC curve. 

Point 1 and 2 should be at least discussed in the limitations.

Author Response

We wish to thank you all for your constructive comments in this round of review. Your comments provided valuable insights to refine its contents and analysis. In this document, we try to address the issues raised as best as possible.

We agree with the reviewer's opinion about the limited ICU care of older patients. Therefore we highlight it in the limitations section including bibliographic references that discussing the importance of this problem.

Line 285: “In the sample distribution according to the need for intensive care of the patients, we found that age groups were significantly different; this situation may be due to confounding bias with older patients who often have limited ICU care [37,38].”

  1. Angus, D.C. Admitting Elderly Patients to the Intensive Care Unit-Is it the Right Decision?. JAMA 2017,318(15),1443–1444.
  2. Haas, L.; de Lange, D.W.; van Dijk, D.; van Delden, J. Should we deny ICU admission to the elderly? Ethical considerations in times of COVID-19. Critical care (London, England) 2020,24(1),321.

We appreciate the reviewer's comment, and we take into account that the pathologies distribution is a factor that supposes important limitations in our work (situation exposed in the limitations section).

As suggested, we include the limitation that 90% of ICU patients present ischaemic stroke, seizures and haemorrhage, proposing future studies in this line related to the predictive capacity of EWS in “unplanned ICU-admission”, or also called “indirect ICU admission”, as well as in “direct ICU admission from the EDs”. We added the following paragraph:

Line 293-297: “90% of the patients admitted to the ICU presented as a base pathology ischaemic stroke, seizures or haemorrhage. We consider interesting to carry out future studies to address the predictive capacity of the EWS, discriminating between “unplanned ICU-admission”, as well as “direct ICU admission from the EDs” in patients with these pathologies”.

We agree with the reviewer. We moved table 4 to supplementary materials. We also attached in the present letter the figure showing the ROC curves related to table 3. We decided for simplicity reasons, not to include it in the manuscript.

Round 2

Reviewer 2 Report

The requested changes were not taken into account.